# A Survey on Treatment Status of Korean Medicine Doctors for the Behavioral and Psychological Symptoms of Dementia: Preliminary Data for Guidance of Integrative Care

**DOI:** 10.3390/healthcare10020269

**Published:** 2022-01-29

**Authors:** Chan-Young Kwon, Boram Lee

**Affiliations:** 1Department of Oriental Neuropsychiatry, Dong-Eui University College of Korean Medicine, 52-57, Yangjeong ro, Busanjin-gu, Busan 47227, Korea; 2KM Science Research Division, Korea Institute of Oriental Medicine, 1672 Yuseong-daero, Yuseong-gu, Daejeon 34054, Korea; qhfka9357@naver.com

**Keywords:** dementia, BPSD, Korean medicine, integrative medicine, survey

## Abstract

Behavioral and psychological symptoms of dementia (BPSD) are major causes of care burden in patients with dementia. Integrative care, including Western medicine (WM) and Korean medicine (KM), can be an attractive option for this condition. To investigate the experience and perceptions of dementia care; experience, knowledge, and perceptions of management for BPSD; the need for guidance of integrative care for BPSD management, KM doctors were surveyed. A total of 137 KM doctors completed the survey. Most institutions where the participants worked were able to provide integrative care services (88.32%). The rate of referral for dementia patients from the WM to KM departments was also highest in the convalescent hospital setting (94.92%), while the rate was lowest in the public health center setting (38.46%). The common purpose of KM treatments for patients with dementia included “psychological symptom treatment” (37.23%); however, the number of referrals from WM to KM departments for BPSD management was relatively few (25.64%). Participants perceived that knowledge about KM or integrative care for BPSD of non-KMD personnel working at the same institution was generally low. Based on these results, facilitating mutual exchange between WM and KM can help establish integrative care for dementia management in Korea.

## 1. Introduction

Dementia is characterized by cognitive impairment that significantly interferes with functions of daily living and is commonly caused by Alzheimer’s disease (AD), vascular dementia (VaD), Lewy body dementia, and frontotemporal dementia. As a leading cause of the disease burden, approximately 50 million people were living with dementia in 2018 worldwide, and this number is expected to reach 152 million by 2050 [1]. In a systematic review, the total annual cost for dementia patients was estimated to be, on average, USD 30,554 per patient [2], and the cost is expected to gradually increase [1]. Although there are some treatable and reversible dementias, including head injury, alcohol-related dementia, hypothyroidism, meningoencephalitis, and neurosyphilis, the disease-modifying treatment of the main causes of dementia, including AD, has not yet been established; therefore, early diagnosis and prevention of the progression of this disorder are of paramount importance [3]. Recent studies have emphasized the multiple etiologies of dementia, particularly the importance of multidisciplinary management and lifestyle modification [3,4,5].

The behavioral and psychological symptoms of dementia (BPSD), including agitation, aggression, apathy, wandering, depression, and sleep disturbance, that occur during the course of dementia render care at home, in nursing homes, and in hospital environments difficult and are major causes of care burden [6]. A recent systematic review indicated that the prevalence of BPSD in community-dwelling patients with dementia varied from 4% to 32% depending on the symptoms, among which include apathy, depression, anxiety, irritability, agitation, aggression, sleep disorders, and eating disorders [7]. BPSD is not usually caused by a single etiology, but rather it is triggered by patient factors, such as unmet needs, pain, comorbidities, and personality, and environmental factors including caregiving quality, life events, caregiver distress, and family dynamics [8]. As recent findings have suggested, there is a relationship between BPSD and neurodegeneration in terms of cerebrospinal fluid tau and brain atrophy, and BPSD may be considered a useful indicator of the pathological progression of dementia [9]. Several clinical practice guidelines emphasize the need to preemptively provide non-pharmacological therapies rather than pharmacological therapies and to limit the use of psychotropic drugs, especially antipsychotics [3,10,11]. It is known that the use of antipsychotics in this population is associated with adverse events such as extrapyramidal symptoms, edema, falls, cardiovascular events, and even death [12,13]. However, in actual clinical practice, antipsychotics, especially atypical antipsychotics, are still commonly used because of the limited availability of non-pharmacological therapies and medical personnel performing them [14].

Integrative medicine can be defined as “practicing medicine in a way that selectively incorporates elements of complementary and alternative medicine (CAM) into comprehensive treatment plans alongside solidly orthodox methods of diagnosis and treatment” [15]. Solving problems that are limited by conventional medicine alone is receiving attention including the use of integrative medicine in the management of dementia patients [16,17,18]. Korea has a dual medical system that includes Western medicine (WM) and Korean medicine (KM), and in the national medical system, patients are guaranteed their own medical options [19]. In particular, at hospitals, convalescent hospitals, or public health centers that provide both WM and KM services, patients with dementia may have the opportunity to receive integrative care. Furthermore, following the recent statute revision in June 2021, KM doctors (KMDs) were included as essential medical staff in dementia safety hospitals for the purpose of intensively managing dementia patients with severe BPSD or delirium in Korea. However, an optimal protocol for integrative medicine, including WM and KM, for patients with dementia has not yet been developed. Although KM and WM are already implemented as integrative medicine for dementia patients in clinical settings in Korea, no survey has focused on integrative medicine for managing BPSD.

Therefore, in this survey, KMDs who currently treat patients with dementia were targeted to investigate the experiences and perceptions of dementia care; experience, knowledge, and perceptions of WM and KM management for BPSD; the need for guidance on integrative medicine for BPSD management in their clinical settings. The results of this survey can be used to develop a guide for medical institutions to effectively apply dementia management to BPSD patients using integrative medicine in Korea.

## 2. Materials and Methods

### 2.1. Participants

A cross-sectional survey was conducted among KMDs who satisfied the following criteria: (1) This survey targeted KMDs working in KM, convalescent, or public health centers. The reason for the limitation of these clinical settings is that this survey was intended to develop guidance on integrative care for patients with dementia, especially for BPSD. Korea has a dual medical system with both WM and KM, and these facilities can provide medical services to patients with dementia at one institution. (2) This survey targeted KMDs who treated at least one dementia patient every week.

### 2.2. Questionnaire Construction

The initial draft of the questionnaire was constructed after a comprehensive literature review by a KMD, who is a KM neuropsychiatric specialist with more than 8 years of clinical experience and an assistant professor at the College of KM in Korea. The KMD is also the director of the KM Dementia Center at the university-affiliated KM hospital. Next, this draft was reviewed and agreed upon by a research team consisting of two professors (majoring in KM neuropsychiatry and Sasang constitution, respectively) with more than 20 years of clinical experience, a KM neuropsychiatric specialist working at a public health center with more than 6 years of clinical experience, and a resident (majoring in KM internal medicine) with more than 2 years of clinical experience. The final version of the questionnaire consisted of (a) the sociodemographic characteristics of the respondents (i.e., sex, age, clinical experience, working area, and KM specialist qualifications); (b) the current status, experience, and perceptions of dementia care; (c) experience, knowledge, and perceptions of WM and KM management for BPSD; (d) the need for guidance of integrative medicine for BPSD management.

### 2.3. Distribution and Collection of Questionnaires

This survey was provided as an anonymous online form using Survey Monkey (Momentive Inc., San Mateo, CA, USA), and a link to this survey was sent to KMDs’ e-mail addresses with the cooperation of the Association of Korean Medicine. In the e-mail, the purpose and content of this survey, rights of the participants, potential rewards, contact information for the researcher, and a link to the survey were provided. Only KMDs who evaluated the detailed explanation of this study and agreed to participate voluntarily were allowed to participate in this survey. The participants were asked to answer a total of 52 questions of which 28 requested responses on a 1–5 Likert scale (1—strongly disagree (or very little) to 5—strongly agree (or very much)). Seven questions permitted multiple responses, and two permitted open responses that were limited to natural numbers. The questionnaire and response data are presented in Appendix A. Considering a previous similar survey [20] and the period of this survey, the total number of expected respondents was 150, and it was announced that 100 respondents who completed the survey would be randomly drawn and a coffee gift card worth KRW 4000 (approximately USD 3.4) would be sent. All relevant questions were set as mandatory, and no missing values occurred, as a non-response did not allow for proceeding to the next step. The survey was conducted from 29 September to 10 November 2021, and invitation emails were sent a total of three times during this period.

### 2.4. Data Entry

After the survey was completed, individual response files without identifying information were downloaded from Survey Monkey (Momentive Inc., San Mateo, CA, USA) as a Microsoft Office Excel 365 (Microsoft Corporation, Redmond, WA, USA) data sheet and were analyzed. Only data from the participants who completed all responses to the last question were analyzed, and the data of participants who did not agree to participate in this survey or who ended the survey without completing the survey were excluded from the analysis.

### 2.5. Statistical Analysis

Collected data were analyzed using descriptive statistics and presented as frequencies (percentages, %). To compare the differences in responses between the KMDs belonging to each clinical setting, categorical variables were compared using the Chi-square test. The normality of continuous variables in the clinical data was tested using the Kolmogorov–Smirnov test. Significant differences between the groups were investigated using one-way analysis of variance (ANOVA). Statistical significance was set at *p*-values < 0.05. In addition, Likert scores or numerical data were expressed as the mean or median value and range. All results were analyzed using Microsoft Office Excel 365 (Microsoft Corporation, Redmond, WA, USA) and SAS^®^ version 9.4 (SAS Institute Inc., Cary, NC, USA).

### 2.6. Ethical Consideration

The study protocols were approved by the Institutional Review Board of the Dongeui University Korean Medicine Hospital (IRB No. DH-2021-03, approved on 31 May 2021). The study was conducted in accordance with the guidelines of the Declaration of Helsinki. At the beginning of the survey, informed consent forms were displayed and only participants who consented though this informed consent form participated in this survey.

## 3. Results

A total of 334 KMDs accessed the online survey, and 162 satisfied the inclusion criteria and initiated the survey. Most of the dropouts occurred because they did not meet the criteria for inclusion in the survey at KMDs’ workplace or the current status of treatment for dementia patients. Finally, 137 participants, 63 from KM hospitals, 61 from convalescent hospitals, and 13 from public health centers, completed the survey. Thus, this survey achieved its target response with a completion rate of 84.57% (137/162) (Figure 1).

### 3.1. Participants’ Characteristics

Among the participants, women accounted for 31.39% (*n* = 43), and those in their 30s (59/137, 43.07%) or 40s (38/137, 27.74%) were common. The most commonly reported clinical experience of KMDs was 5–9 years (44/137, 32.12%), followed by 10–14 years (24/137, 17.52%). Most of the currently working areas were Seoul (43/137, 31.39%) or Gyeonggi-do (29/137, 21.17%), the capital area of Korea. Among the participants, the proportion of licensed specialists was 37.23% (51/137), and KM internal medicine (16/51, 31.37%) and acupuncture and moxibustion (11/51, 21.57%) accounted for approximately half of the specialists. Regarding age (*p* = 0.0002), clinical experience (*p* = 0.0053), and type of KMD license (*p* = 0.0002), there were significant differences between the groups in the Chi-square test (Table 1).

### 3.2. The Current Status and Experience of Integrative Treatment for Dementia Patients

In the total sample, the average number of dementia patients treated per month was five or less per month (72/137, 52.55%), and 21 or more per month (28/137, 20.44%) was common. In particular, in the convalescent hospital setting, KMDs who reported treating more than 21 patients with dementia per month (29/61, 47.54%) were the most frequent. The proportion of inpatients with dementia was most commonly reported to be less than 25% (33/63, 52.38%) in the KM hospital setting but 75% or more (29/61, 47.54%) in the convalescent hospital setting. Additionally, AD was the most common type of dementia in patients with treated dementia (108/137, 78.83%), followed by VD (63/137, 45.99%) and PDD (48/137, 35.04%) (Figure 2).

A difference was observed in the purpose of KM service for patients with dementia, and the most common purpose in the total sample was treatment of cognitive impairment (71/137, 51.82%) and improvement of general condition (69/137, 50.36%). In addition, psychological symptom treatment was the third most common purpose (51/137, 37.23%). In the KM hospital setting, the most common purpose of KM service was to treat cognitive impairment (38/63, 60.32%), whereas in the convalescent hospital setting, the most common purpose was to improve the overall condition (34/61, 55.74%). In the public health center setting, the two purposes were the highest at the same rate (7/13, 53.85%) (Figure 3a). In all three settings, acupuncture was the main KM treatment used for dementia patients (130/137, 94.89%), but the second most frequent was uninsured herbal medicine (decoction) (39/63, 61.90%), moxibustion (35/61, 57.38%), and insured herbal medicine (extract) (7/13, 53.85%) in KM hospital, convalescent hospital, and public health center settings, respectively (Figure 4a). There was also a difference with respect to the cost of the service. In the KM hospital setting, the rate of response with average treatment cost (out-of-pocket) per treatment between KRW 10,000 and less than KRW 20,000 was the highest (27/63, 42.86%), whereas in convalescent hospital, and public health center setting, less than KRW 5000 was the most common (27/61, 44.26%; 7/13, 53.85%). Regarding the service cost, a significant difference was confirmed between the three groups using the Chi-square test (*p* < 0.0001) (Table 2).

In settings where most of the respondents worked, integrative care was being implemented in the institution (121/137, 88.32%). Among them, the rate of integrative care currently implemented for dementia patients was reported to be over 50% in 49.59% (60/121), and in the convalescent hospital setting, 54.24% (32/59) reported that they were providing integrative care for dementia patients in more than 75% of cases. The rate of referral for dementia patients from the WM department to KM departments was different according to the clinical setting, which was 74.38% in the total sample, 59.18% (29/49) in the KM hospital settings, 94.92% (56/59) in the convalescent hospital setting, and 38.46% (5/13) in the public health center setting. The reason for referral did not differ from the reason for KM treatment, namely, treatment of cognitive impairment (14/29, 48.28%) in the KM hospital setting, whereas in the convalescent hospital setting, the most common purpose was to improve the overall condition (39/56, 69.64%). In the public health center setting, the two purposes were the highest at the same rate (2/5, 40%) (Figure 3b). The rate of referral for dementia patients from the KM department to WM departments was 64.46% in the total sample: 81.63% (40/49) in the KM hospital settings, 54.24% (32/59) in the convalescent hospital setting, and 46.15% (6/13) in the public health center setting. In all settings, the most frequent reason for referral was WM assessment and evaluation for dementia in the total sample (43/78, 55.13%), in the KM hospital settings (22/40, 55%), in the convalescent hospital setting (16/32, 50%), and in the public health center setting (5/6, 83.33%) (Figure 3c).

### 3.3. The Perception of BPSD Management

Participants generally thought that KM treatment (mean score on a 1–5 Likert scale: in a total sample, 3.80; in the KM hospital setting, 3.78; in the convalescent hospital setting, 3.70; in the public health center setting, 4.31), WM treatment (3.53, 3.51, 3.54, and 3.54), and integrative care (3.98, 3.92, 3.95, and 4.38) were effective in BPSD management. They also reported that KM treatment (4.10, 4.05, 4.08, and 4.46), WM treatment (3.31, 3.41, 3.26, and 3), and integrative care (3.96, 3.94, 3.97, and 4.08) were safe for BPSD management. Regarding the safety of integrative care for BPSD (*p* = 0.0256), there was a significant difference between the groups following the Chi-square test (Table 3). The most effective KM treatment for BPSD management was acupuncture in the total sample (117/137, 85.40%), KM hospital setting (55/63, 87.60%), and convalescent hospital setting (53/61, 86.89%). However, uninsured herbal medicine (decoction) was frequently reported to be the most effective in the public health center setting (12/13, 92.31%) (Figure 4b). The median number of minimum treatment days required for BPSD management (i.e., the minimum number of days required to obtain a therapeutic effect) was the same for all settings (30 (range, 2–365), 30 (2–180), 30 (3–365), 30 (3–180)). In addition, the median number of the maximum number of treatment days for BPSD management (i.e., the maximum number of days that can be attempted even if the treatment effect is not obtained) was also the same in all settings (90 (5–9999), 90 (6–2000), 90 (5–7300), 90 (6–9999)). ANOVA did not reveal any significant differences between the three groups for these two questions (both *p* > 0.05). Most of the participants advocated the need for integrative care for BPSD (129/137, 94.16%), and the most common reasons answered included the following: integrative care increases patient reliability and compliance (72/129, 55.81%), patients and caregivers preferred integrative care (62/129, 48.06%), and it is empirically supported that integrative care is effective in managing BPSD (57/129, 44.19%) (Figure 5).

### 3.4. The Knowledge of BPSD Management

Most scores on questions about knowledge about BPSD were between 3 (neutral) and 4 (slightly much) on a 1–5 Likert scale: definition of BPSD (3.26 in the total sample, 3.33 in the KM hospital setting, 3.08 in the convalescent hospital setting, and 3.69 in the public health center setting), assessment method of BPSD (3.11, 3.21, 2.85, and 3.85), differences in BPSD by type of dementia (2.94, 3.08, 2.72, and 3.31), differences in BPSD by severity of dementia (3.04, 3.10, 2.87, and 3.54), effectiveness of WM treatment on BPSD (3.14, 3.22, 2.98, and 3.46), safety of WM treatment on BPSD (3.04, 3.13, 2.87, and 3.46), effectiveness of KM treatment on BPSD (3.47, 3.54, 3.33, and 3.85), safety of KM treatment on BPSD (3.59, 3.63, 3.48, and 3.92), effectiveness of integrative care on BPSD (3.40, 3.37, 3.38, and 3.69), and safety of integrative care on BPSD (3.40, 3.38, 3.38, and 3.62). Although the differences were not significant, the mean response scores of participants in the convalescent hospital setting were generally lower than those in the other two settings. In most cases, participants in the public health center setting had the highest response scores. Meanwhile, knowledge of KM or integrative care for BPSD of non-KMD personnel in the same institution was evaluated to be lower, and it was usually distributed between points 2 (slightly less) and 3 (neutral): effectiveness of KM treatment on BPSD (2.80, 2.81, 2.72, an 3.15), safety of KM treatment on BPSD (2.88, 2.92, 2.77, and 3.15), effectiveness of integrative care on BPSD (2.78, 2.79, 2.75, and 2.85), and safety of integrative care on BPSD (2.83, 2.84, 2.82, and 2.85). Regarding the BPSD assessment method (*p* = 0.0134), there was a significant difference between the groups in the Chi-square test (Table 3).

### 3.5. Perceptions and Needs of Manuals to Promote Integrative Care on BPSD

Most of the participants responded that manuals to promote integrative care on BPSD were necessary (128/137, 93.43%). Most scores on questions about the need for the contents were between 4 (agree) and 5 (strongly agree) on a 1–5 Likert scale: knowledge of BPSD (definition, types, and assessment) (4.09 in the total sample, 4.17 in the KM hospital setting, 4.02 in the convalescent hospital setting, 4 in the public health center setting), evidence of WM treatment on BPSD (3.95, 4.07, 3.83, and 3.91), evidence of KM treatment on BPSD (4.19, 4.24, 4.10, and 4.36), evidence of integrative care on BPSD (4.09, 4.12, 4.02, and 4.27), criteria for KM treatment referral (4.01, 4.03, 3.92, and 4.36), standardized referral form for KM treatment (3.94, 3.98, 3.85, and 4.18), standardized reply form of KM treatment (4.01, 4, 3.98, and 4.18), and scenario of integrative medicine for BPSD (4.03, 4.10, 4.02, and 3.73). There was no significant difference among the three groups in the Chi-square test (all *p* > 0.05) (Table 4).

## 4. Discussion

### 4.1. Main Findings of the Survey

According to survey responses obtained from 137 KMDs who were currently treating patients with dementia, KMDs working in a convalescent hospital setting treated dementia patients most frequently (21 patients or more per month, 47.54%) and had the highest proportion of inpatients with dementia (75% or more, 47.54%). As expected, AD was the most common type of dementia that was treated with KMD. Most of the institutions where the participants worked were able to provide integrative care services (88.32%), and half of the participants (49.59%) reported that integrative care is currently provided for more than 50% of their dementia patients. More than half of KMDs (54.24%) working in convalescent hospital settings reported that integrative care is currently provided for more than 75% of patients with dementia. The rate of referral for dementia patients from the WM department to KM departments was also highest in the convalescent hospital setting (94.92%), while the rate was lowest in the public health center setting (38.46%). The common purpose of KM treatments for patients with dementia included “treatment of cognitive impairment” (51.82%), “improvement of the general condition” (50.36%), and “psychological symptom treatment” (37.23%). The common reason for the referral from the WM department to KM departments for patients with dementia included “improvement of the general condition” (58.89%), “treatment of cognitive impairment” (41.11%), and “treatment of non-dementia condition” (37.78%). Finally, the common referral reason from the KM department to WM departments for patients with dementia included “assessment of dementia” (55.13%), “assessment of non-dementia conditions” (41.03%), and “treatment of non-dementia conditions” (35.90%). The most commonly used KM treatment for patients with dementia was acupuncture (94.89% of the total sample), followed by uninsured herbal medicine (decoction), moxibustion, and insured herbal medicine (extract), according to the clinical settings. In addition, the most effective KM treatment for BPSD was acupuncture (85.40%) as indicated by the participants. There was a difference with respect to the cost of the KM service, and in KM hospital setting, the most commonly reported response was that the average treatment cost (out-of-pocket) per treatment was between KRW 10,000 and less than KRW 20,000 (42.86%), and in the other two clinical settings, the most commonly reported response was less than KRW 5000 (44.26% in the convalescent hospital setting; 53.85% in the public health center setting) (Chi-square test, *p* < 0.0001). The participants’ perceptions (advocacy) about the effectiveness and safety of WM, KM, and integrative care in managing BPSD were generally high, and the level of knowledge about BPSD was also generally high. However, the participants evaluated that non-KMD personnel working at the same institution had a low level of knowledge about KM or integrative care for BPSD. Most of the participants supported the need for manuals to promote integrative care on BPSD (93.43%).

### 4.2. Clinical Interpretation

Dementia is the main cause of disease burden worldwide [1], and BPSD is associated with the burden of disease and care [6]. Multidisciplinary and individualized management is prioritized for the management of dementia patients [3,4,5], and integrative medicine that provides individualized management from a holistic concept may be a promising strategy for dementia management. However, a framework for providing integrative medicine in the clinical field is still lacking. Moreover, since Korea has a dual medical system of WM and KM, it is important to investigate the awareness and cooperation status of medical personnel on integrative medicine for patients with dementia to establish an integrative dementia care strategy.

According to the results of this survey, convalescent hospital settings appear to be the environment in which integrated medical services for dementia patients are frequently provided. However, an integrated care manual has not yet been developed in Korea for the management of dementia patients or elderly patients in convalescent hospital settings. Moreover, according to a 2014 survey, most of the cooperation between WM and KM in this environment lacks mutual exchange [21]. The purpose of KMD’s treatment for dementia and the treatment of dementia patients referred by the WM departments included “treatment of cognitive impairment” and “improvement of the general condition”. The results indicate that KM treatment, such as acupuncture, is being attempted as an additional option for improving cognitive impairment where conventional medicine has limitations [3]. In addition, it seems to reflect the recognition and evidence that KM treatments, such as acupuncture and herbal medicine, can improve the overall condition of an individual from a holistic concept [22]. However, while KMD ranked third for the psychological symptom treatment for the purpose of dementia treatment, referral from WM departments to KM departments for BPSD management was lacking. Participants perceived that knowledge about KM or integrative care for BPSD of non-KMD personnel working at the same institution was generally low, and the low knowledge may have led to a decline in the use of KM treatment or integrative care for BPSD management. Most of the participants advocated for the development of a manual to promote cooperative medical care with non-KMD personnel in dementia care, and the future-developed manual will raise awareness regarding KM treatment for BPSD management and potentially help establish integrative medical care for dementia patients. Considering the limited resources for development, convalescent hospital settings have the highest priority for the development of this manual.

### 4.3. Results of Previous Studies

Worldwide surveys focused on integrative medicine for BPSD have not been conducted; however, some related studies can be referenced. A US cross-sectional survey in 2007 revealed that 62.9% of community-dwelling older adults reported using one or more CAM modalities [23]. Similar results were obtained in a German cross-sectional survey in 2014, confirming that 61.3% of the elderly aged ≥70 years used CAM modalities [24]. When the purpose is limited to dementia, the use rate is estimated to be 18.4%, but the frequency of CAM use remains high [25]. The use of CAM in dementia patients and their caregivers is widespread, and they have an overall positive perception of the use of CAM [26]. Moreover, a qualitative study conducted on the German elderly found that most participants welcomed the incorporation of CAM into their health care [27]. However, few studies have examined the knowledge and attitudes of clinicians regarding the use of integrative medicine for dementia management. Moreover, as many CAM users do not inform their general practitioners of their CAM use [23,24], efforts to improve the knowledge and attitudes of integrative medicine from the perspective of health care providers in the provision of geriatric care, including dementia, are required.

### 4.4. Limitations

This study had several limitations. First, the sample of KMDs working in public health center settings was small with only 13 people; therefore, the answers could not be representative of the entire KMD in the public health center settings. However, according to a survey by the Korean Statistical Information Service, as of 2021, the proportion of KMDs working in public health center settings was only 4.06% (916/22,584) [28], which seems to be an unavoidable result. Therefore, if a qualitative study is conducted on KMDs working in public health center settings in the future, it is possible to obtain the results of supplementing insufficient quantitative data. Second, the knowledge level of the BPSD of non-KMD personnel is based on the opinions of the KMDs rather than directly asking them; therefore, this should be used only as a reference. Finally, recall bias is likely, because the survey depended on the memory of the respondents. Nevertheless, this is the first study to understand the current status of KMDs’ experiences, perceptions, and knowledge about the treatment of BPSD patients in various clinical environments. In the absence of related research, it can provide basic data for developing an integrative medicine manual according to the different clinical environments for the treatment of BPSD in the future by identifying the status of KM treatment in real-world clinical settings.

## 5. Conclusions

This was the first survey to investigate the experiences and perceptions of dementia care; experience, knowledge, and perception of WM and KM management for BPSD; the need for guidance of integrative medicine for BPSD management in their clinical settings. Based on the results, facilitating a mutual exchange between WM and KM can help establish integrative care for dementia management in Korea. In particular, from the perspective of integrative care, the use of KM treatment for BPSD management is under-recognized by non-KMD personnel; therefore, it is necessary to conduct research investigating the pros and cons of introducing KM treatment in BPSD management and to promote and educate the results.

Suggestions for further studies include the following: A survey and qualitative research on the experience, perception, and knowledge of KM treatment and integrative care for BPSD patients or caregivers that can be conducted, which will help reflect the patient’s perspective in the manual’s development of future integrative medicine. Therefore, the perception, knowledge, and experience of KM treatment of non-KMD personnel working in various clinical environments can also be investigated where integrative care for BPSD is possible. Furthermore, clinical pathways can be developed for each clinical environment based on the manual to be developed in accordance with this study. Evaluating the effectiveness, safety, and satisfaction of medical staff and patients through observational studies is also necessary when these pathways are applied. Furthermore, since various herb–drug interactions have been reported [29], it is necessary to systematically monitor safety reports via a prospective registry when administering integrative treatment for dementia patients.

## Figures and Tables

**Figure 1 healthcare-10-00269-f001:**
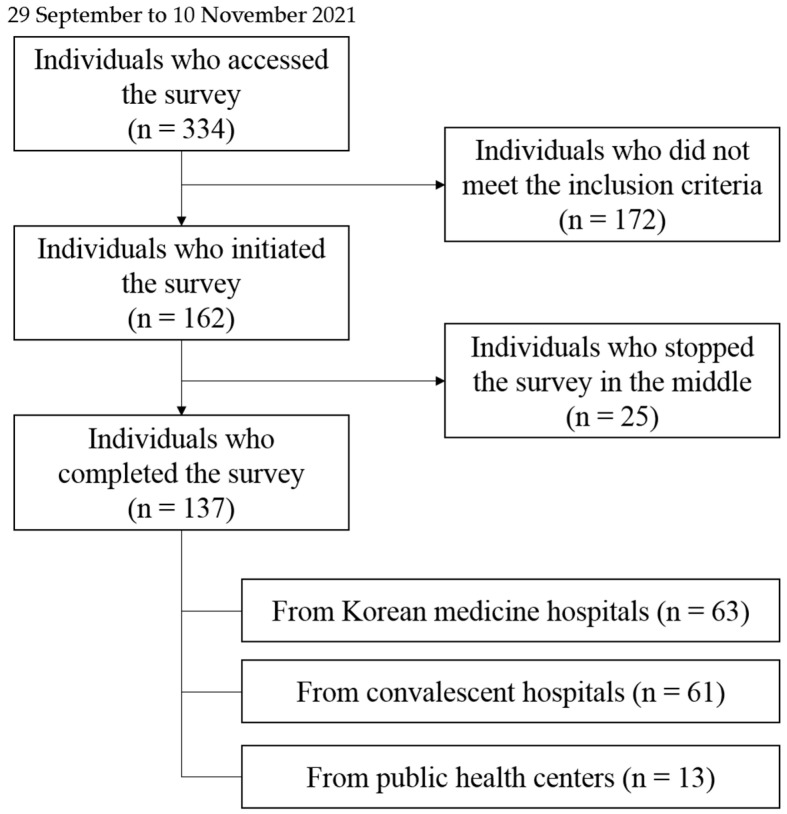
Flow chart of this survey.

**Figure 2 healthcare-10-00269-f002:**
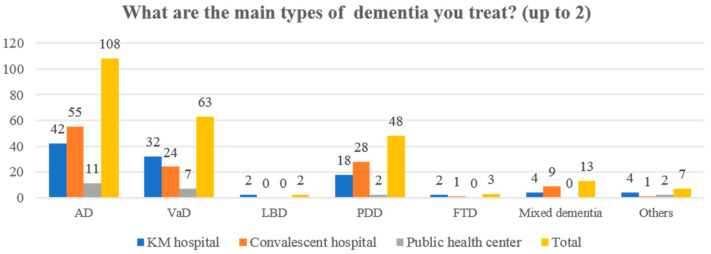
Types of dementia treated by participants (*n* = 137). AD, Alzheimer’s disease; FTD, frontotemporal dementia; KM, Korean medicine; LBD, Lewy body dementia; PDD, Parkinson’s disease dementia; VaD, vascular dementia.

**Figure 3 healthcare-10-00269-f003:**
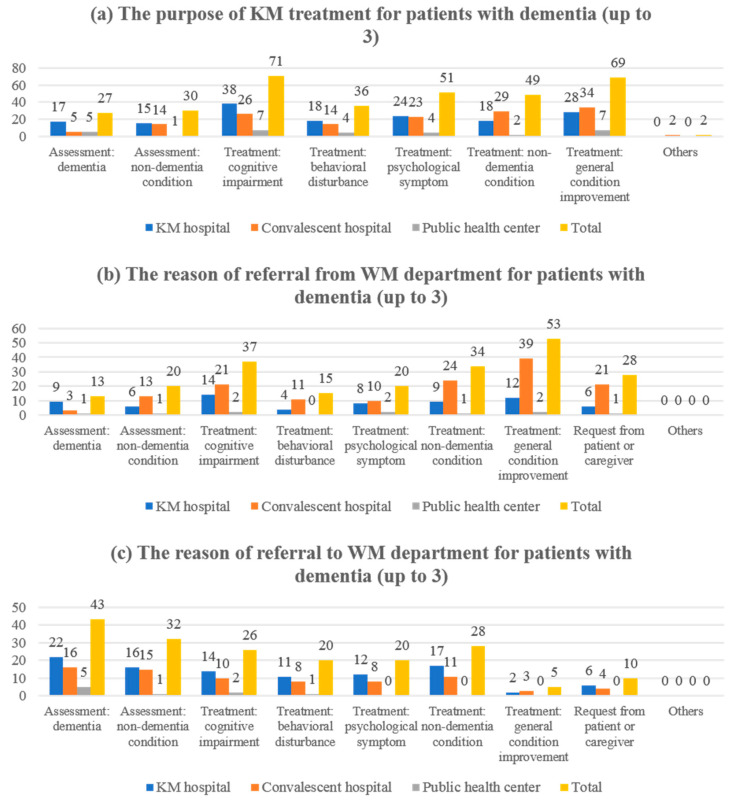
The purpose of treatments for patients with dementia: (**a**) KM treatment for patients with dementia (*n* = 137); (**b**) reason for referral from WM department for patients with dementia (*n* = 85); (**c**) reason for referral to WM department for patients with dementia (*n* = 73). KM, Korean medicine; WM, Western medicine.

**Figure 4 healthcare-10-00269-f004:**
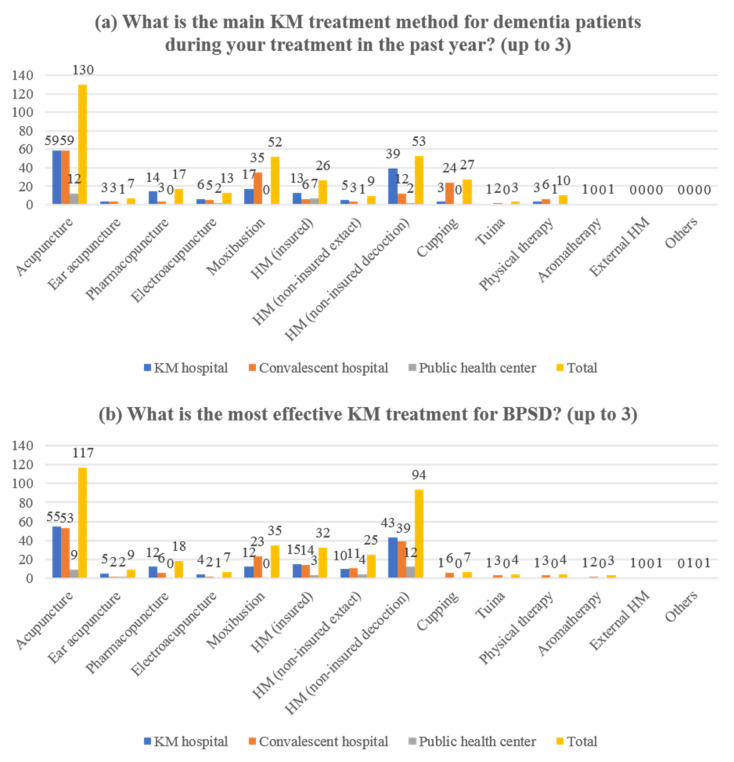
KM treatments for patients with dementia and BPSD: (**a**) use of KM treatment for patients with dementia (*n* = 137); (**b**) the perception of effectiveness of KM treatment for BPSD (*n* = 137). BPSD, behavioral and psychological symptoms of dementia; HM, herbal medicine; KM, Korean medicine; WM, Western medicine.

**Figure 5 healthcare-10-00269-f005:**
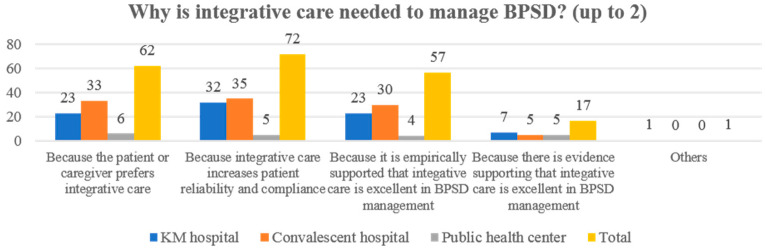
The reason for the need of integrative care for BPSD (*n* = 129). BPSD, behavioral and psychological symptoms of dementia.

**Table 1 healthcare-10-00269-t001:** Sociodemographic characteristics of respondents.

Classification	KM Hospital (*n* = 63) (*n* (%))	Convalescent Hospital (*n* = 61) (*n* (%))	Public Health Center (*n* = 13) (*n* (%))	Total (*n* = 137) (*n* (%))
Sex				
Male	40 (63.49%)	42 (68.85%)	12 (92.31%)	94 (68.61%)
Female	23 (36.51%)	19 (31.15%)	1 (7.69%)	43 (31.39%)
Age (years) ^¶^				
<30	9 (14.29%)	2 (3.28%)	6 (46.15%)	17 (12.41%)
≥30 to <40	30 (47.62%)	22 (36.07%)	7 (53.85%)	59 (43.07%)
≥40 to <49	14 (22.22%)	24 (39.34%)	0 (0%)	38 (27.74%)
≥50 to <59	10 (15.87%)	11 (18.03%)	0 (0%)	21 (15.33%)
≥60	0 (0%)	2 (3.28%)	0 (0%)	2 (1.46%)
Clinical experience (years) ^¶^				
<5	10 (15.87%)	5 (8.20%)	6 (46.15%)	21 (15.33%)
≥ 5 to <10	20 (31.75%)	17 (27.87%)	7 (53.85%)	44 (32.12%)
≥10 to <15	13 (20.63%)	11 (18.03%)	0 (0%)	24 (17.52%)
≥15 to <20	7 (11.11%)	14 (22.95%)	0 (0%)	21 (15.33%)
≥20 to <30	9 (14.29%)	6 (9.84%)	0 (0%)	15 (10.95%)
≥30	4 (6.35%)	8 (13.11%)	0 (0%)	12 (8.76%)
Type of KM doctor license ^¶^				
Specialist	35 (55.56%)	12 (19.67%)	4 (30.77%)	51 (37.23%)
General practitioner	28 (44.44%)	49 (80.33%)	9 (69.23%)	86 (62.77%)

KM, Korean medicine. ^¶^ A significant difference was confirmed between the three groups using the Chi-square test.

**Table 2 healthcare-10-00269-t002:** Cost of KM treatment in patients with dementia.

Classification	KM Hospital (*n* = 63) (*n* (%))	Convalescent Hospital (*n* = 61) (*n* (%))	Public Health Center (*n* = 13) (*n* (%))	Total (*n* = 137) (*n* (%))
Average treatment cost (out-of-pocket) per treatment (KRW) ^¶^				
<5000	5 (7.94%)	27 (44.26%)	7 (53.85%)	39 (28.47%)
≥5000 to <10,000	12 (19.05%)	10 (16.39%)	5 (38.46%)	27 (19.71%)
≥10,000 to <20,000	27 (42.86%)	16 (26.23%)	1 (7.69%)	44 (32.12%)
≥20,000 to <50,000	15 (23.81%)	8 (13.11%)	0 (0%)	23 (16.79%)
≥50,000 to <100,000	3 (4.76%)	0 (0%)	0 (0%)	3 (2.19%)
≥100,000	1 (1.59%)	0 (0%)	0 (0%)	1 (0.73%)

KM, Korean medicine; KRW, Korean Won. Note that USD 1 = KRW 1192 (21 December 2021). ^¶^ A significant difference was confirmed between the three groups using the Chi-square test.

**Table 3 healthcare-10-00269-t003:** The perceptions and knowledge of BPSD management.

Classification Likert Scale (1 (Strongly Disagree (or Very Little) to 5 Strongly Agree (or Very Much))	KM Hospital (*n* = 63) (Mean ± SD)	Convalescent Hospital (*n* = 61) (Mean ± SD)	Public Health Center (*n* = 13) (Mean ± SD)	Total (*n* = 137) (Mean ± SD)
Perceptions (advocacy) of effectiveness				
WM treatments for BPSD	3.51 ± 0.77	3.54 ± 0.64	3.54 ± 1.08	3.53 ± 0.76
KM treatments for BPSD	3.78 ± 0.68	3.70 ± 0.66	4.31 ± 0.61	3.80 ± 0.68
Integrative care for BPSD	3.92 ± 0.72	3.95 ± 0.73	4.38 ± 0.62	3.98 ± 0.73
Perceptions (advocacy) of safety				
WM treatments for BPSD	3.41 ± 0.85	3.26 ± 0.74	3 ± 1.11	3.31 ± 0.84
KM treatments for BPSD	4.05 ± 0.79	4.08 ± 0.63	4.46 ± 0.63	4.10 ± 0.72
Integrative care for BPSD ^¶^	3.94 ± 0.66	3.97 ± 0.68	4.08 ± 0.92	3.96 ± 0.70
Knowledge of BPSD				
Definition of BPSD	3.33 ± 0.93	3.08 ± 0.66	3.69 ± 0.72	3.26 ± 0.82
Assessment method of BPSD ^¶^	3.21 ± 0.86	2.85 ± 0.72	3.85 ± 0.66	3.12 ± 0.83
Differences in BPSD by type of dementia	3.08 ± 0.88	2.72 ± 0.83	3.31 ± 0.72	2.94 ± 0.87
Differences in BPSD by severity of dementia	3.10 ± 0.79	2.87 ± 0.82	3.54 ± 0.93	3.04 ± 0.84
Effectiveness of WM treatment on BPSD	3.22 ± 0.86	2.98 ± 0.74	3.46 ± 0.75	3.14 ± 0.81
Effectiveness of KM treatment on BPSD	3.54 ± 0.83	3.33 ± 0.62	3.85 ± 0.86	3.47 ± 0.76
Effectiveness of integrative care on BPSD	3.37 ± 0.74	3.38 ± 0.68	3.69 ± 0.72	3.40 ± 0.72
Safety of WM treatment on BPSD	3.13 ± 0.83	2.87 ± 0.64	3.46 ± 0.84	3.04 ± 0.77
Safety of KM treatment on BPSD	3.63 ± 0.84	3.48 ± 0.64	3.92 ± 0.73	3.59 ± 0.76
Safety of integrative care on BPSD	3.38 ± 0.76	3.38 ± 0.68	3.62 ± 0.84	3.40 ± 0.74
Knowledge of BPSD (non-KMD personnel)				
Effectiveness of KM treatment on BPSD	2.81 ± 0.99	2.72 ± 0.94	3.15 ± 1.17	2.80 ± 1.00
Effectiveness of integrative care on BPSD	2.79 ± 0.99	2.75 ± 0.90	2.85 ± 1.23	2.78 ± 0.98
Safety of KM treatment on BPSD	2.92 ± 1.07	2.77 ± 0.95	3.15 ± 1.23	2.88 ± 1.04
Safety of integrative care on BPSD	2.84 ± 1.04	2.82 ± 0.97	2.85 ± 1.17	2.83 ± 1.02

BPSD, behavioral and psychological symptoms of dementia; KM, Korean medicine; SD, standard deviation; WM, Western medicine. ^¶^ A significant difference was confirmed between the three groups using the Chi-square test.

**Table 4 healthcare-10-00269-t004:** The perceptions and needs of manuals to promote integrative care on BPSD.

Classification Likert Scale 1 (Strongly Disagree to 5 Strongly Agree)	KM Hospital (*n* = 58) (Mean ± SD)	Convalescent Hospital (*n* = 59) (Mean ± SD)	Public Health Center (*n* = 11) (Mean ± SD)	Total (*n* = 128) (Mean ± SD)
Knowledge of BPSD (definition, types, and assessment)	4.17 ± 0.56	4.02 ± 0.75	4 ± 0.60	4.09 ± 0.66
Evidence of WM treatment on BPSD	4.07 ± 0.72	3.83 ± 0.74	3.91 ± 0.51	3.95 ± 0.72
Evidence of KM treatment on BPSD	4.24 ± 0.62	4.10 ± 0.80	4.36 ± 0.88	4.19 ± 0.74
Evidence of integrative care on BPSD	4.12 ± 0.65	4.02 ± 0.70	4.27 ± 0.75	4.09 ± 0.68
Criteria for KM treatment referral	4.03 ± 0.81	3.92 ± 0.74	4.36 ± 0.77	4.01 ± 0.79
Standardized referral form for KM treatment	3.98 ± 0.71	3.85 ± 0.75	4.18 ± 0.83	3.94 ± 0.75
Standardized reply form of KM treatment	3.98 ± 0.67	3.98 ± 0.72	4.18 ± 0.83	4.01 ± 0.71
Scenario of integrative medicine for BPSD	4.10 ± 0.69	4.10 ± 0.72	3.73 ± 0.86	4.03 ± 0.73

BPSD, behavioral and psychological symptoms of dementia; KM, Korean medicine; SD, standard deviation; WM, Western medicine. There was no significant difference among the three groups in the Chi-square test.

## Data Availability

The datasets used or analyzed during the current study are available from the corresponding author upon reasonable request.

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
