# Peer review of "A Survey on Treatment Status of Korean Medicine Doctors for the Behavioral and Psychological Symptoms of Dementia: Preliminary Data for Guidance of Integrative Care"

_healthcare, 2022, doi:10.3390/healthcare10020269_

Round 1

Reviewer 1 Report

Behavioral and psychological symptoms of dementia (BPSD) are major causes of care  burden in patients with dementia. Integrative care, including Western medicine (WM) and Korean  medicine (KM), can be an attractive option for this condition. The authors proposed a survey to investigate  the experience and perception of dementia care, experience, knowledge, and perception of management for BPSD and  the need for guidance of integrative care for BPSD management. The survey was submitted to  KM doctors. Participants perceived that knowledge about KM or integrative care for BPSD of non-KMD personnel working at the same institution was generally low. Based on the results, facilitating mutual exchange between WM and KM can help establish integrative care for dementia management in Korea.

The contribute is both innovative and interesting.

The following changes are suggested

1) The purpose "Therefore, in this survey, KM doctors (KMDs), who currently treat patients with de mentia, were targeted to investigate the experiences and perceptions of dementia care, experience, knowledge, and perception of WM and KM management for BPSD , and the need for guidance of integrative medicine for BPSD management in their clinical settings "in the introduction must be more explicit and detailed. You should explain the issue you want to address and why. Then later you can explain that you intend to use a survey.

2) You should briefly explain the structure and which modules (graded, choice, multiple choice, likert, open, etc.) you used in the survey in a paragraph before paragraph “2.3. Distribution and collection of questionnaires "perhaps with the title" 2.3 architecture of the survey in brief ". For example, you should say what kind of score there is on the Likert (5 levels appear in the results).

3) You should report an example of the survey (link and print screen) so the reader reads with more motivation and the reviewers understand better. I don't know if it's in the complementary material you've submitted.

4) In the paragraph on statistical analysis you report that the results are translated into percentages (essentially frequencies). Have you used a significance test (Chi square for example which is ideal for frequency testing)? In my opinion it would add value to the study.

5) A flow chart would improve the presentation of the methodology and the interpretation of the results.

6) In the results, the figures must be improved in resolution and where necessary, if they deal with graded or likert questions, the error bars must be placed.

7) Always in the results insert a short paragraph on how the presentation develops.

8) I really like the discussion and the conclusions. Perhaps the “Suggestions for further studies” section I would put it in the conclusions. But that's okay too.

Thank you very much for the opportunity to review an interesting article.

Author Response

Please see the attachment, thank you!

Reviewer 2 Report

The authors report a detailed survey about the current implementation of Western and Korean Medicines in the Health System in Korea. Their findings suggest a potential and beneficial improvement derived from the integration between the two medical approaches and represent an interesting overview of the current therapeutic management of BPSD in Korea, based on which they could design new treatment perspectives.

Some minor revisions:

1) Introduction: BPSD were also investigated based on fluid and imaging biomarkers, and this information can be useful to approach this kind of disorder (see DOI: 10.3389/fnagi.2021.595758 and add this refernce within lines 37-41)

2) Results: I suggest to improve the results descriptions with comparisons and p-value, where possible

3)Discussion: Please, compare the findings of your work with similar survey carried out in other countries, e.g. Europe and USA

Author Response

  • Response to Comments from Reviewer 2

Overall comment:

The authors report a detailed survey about the current implementation of Western and Korean Medicines in the Health System in Korea. Their findings suggest a potential and beneficial improvement derived from the integration between the two medical approaches and represent an interesting overview of the current therapeutic management of BPSD in Korea, based on which they could design new treatment perspectives.

Response:              

Thank you very much for your dedicated review. All changed or added texts in this revised manuscript have been edited in English by a professional proofreading company.

Comment 1:

Some minor revisions:

1) Introduction: BPSD were also investigated based on fluid and imaging biomarkers, and this information can be useful to approach this kind of disorder (see DOI: 10.3389/fnagi.2021.595758 and add this refernce within lines 37-41)

Response 1:           

Thank you for your comments. Based on the reviewer's comments, we added the following sentences to this revised manuscript.

“BPSD is usually not caused by a single etiology, but rather is triggered by patient factors such as unmet needs, pain, comorbidities, and personality, and environmental factors such as caregiving quality, life events, caregiver distress, and family dynamics [8]. As recent findings have suggested a relationship between BPSD and neurodegeneration in terms of cerebrospinal fluid tau and brain atrophy, BPSD may be considered a useful indicator of the pathological progression of dementia [9].” (Please see page 2, red words)

Comment 2:

2) Results: I suggest to improve the results descriptions with comparisons and p-value, where possible

Response 2:           

Thank you for your comments. As advised by the reviewer, we conducted a significance test to analyze the differences in outcomes by clinical settings. Some results have been described in the main text, full results are presented in Supplementary 1.

“Collected data were analyzed using descriptive statistics and presented as frequencies (percentages, %). To compare the differences in responses between the KMDs belonging to each clinical setting, categorical variables were compared using the Chi-square test. The normality of continuous variables in the clinical data was tested using the Kolmogorov-Smirnov test. Significant differences between the groups were investigated using a one-way analysis of variance (ANOVA). Statistical significance was set at ?-values < 0.05. In addition, Likert scores or numerical data were expressed as mean or median value and range. All results were analyzed using Microsoft Office Excel 365 (Microsoft Corporation, Redmond, WA, USA) and SAS® version 9.4 (SAS Institute. Inc., Cary, NC, USA).” (Please see page 4, red words)

“Among the participants, women accounted for 31.39% (n=43), and those in their 30s (59/137, 43.07%) or 40s (38/137, 27.74%) were common. The most commonly reported clinical experience of KMD was 5 to 9 years (44/137, 32.12%), followed by 10 to 14 years (24/137, 17.52%). Most of the currently working areas were Seoul (43/137, 31.39%) or Gyeonggi-do (29/137, 21.17%), the capital area of Korea. Among the participants, the proportion of licensed specialists was 37.23% (51/137), and KM internal medicine (16/51, 31.37%) and acupuncture and moxibustion (11/51, 21.57%) accounted for about half of the specialists. Regarding age (p = 0.0002), clinical experience (p = 0.0053), and type of KMD license (p = 0.0002), there were significant differences between the groups in the Chi-square test (Table 1).” (Please see page 5, red words)

“There was also a difference with respect to the cost of the service. In the KM hospital setting, the rate of response with average treatment cost (out-of-pocket) per treatment between 10,000 KRW and less than 20,000 KRW was the highest (27/63, 42.86%), whereas in convalescent hospital, and public health center setting, less than 5,000 KRW was the most common (27/61, 44.26%; 7/13, 53.85%). Regarding the service cost, a significant difference was confirmed between the three groups using the Chi-square test (p < 0.0001) (Table 2).” (Please see page 7, red words)

“Participants generally thought that KM treatment (mean score on a 1 to 5 Likert scale: in a total sample, 3.80; in the KM hospital setting, 3.78; in the convalescent hospital setting, 3.70; in the public health center setting, 4.31), WM treatment (3.53, 3.51, 3.54, and 3.54), and integrative care (3.98, 3.92, 3.95, and 4.38) were effective in BPSD management. They also reported that KM treatment (4.10, 4.05, 4.08, 4.46), WM treatment (3.31, 3.41, 3.26, 3), and integrative care (3.96, 3.94, 3.97, 4.08) were safe for BPSD management. Regarding the safety of integrative care for BPSD (p = 0.0256), there was a significant difference between the groups following the Chi-square test (Table 3). … In addition, the median number of the maximum number of treatment days for BPSD management (i.e., the maximum number of days that can be attempted even if the treatment effect is not obtained) was also the same in all settings (90 [5–9999], 90 [6–2000], 90 [5–7300], 90 [6–9999]). ANOVA did not reveal any significant differences between the three groups for these two questions (both p > 0.05).” (Please see page 9, red words)

“Meanwhile, the knowledge of KM or integrative care for BPSD of non-KMD personnel in the same institution was evaluated to be lower, and it was usually distributed between points 2 (slightly less) and 3 (neutral): effectiveness of KM treatment on BPSD (2.80, 2.81, 2.72, 3.15), safety of KM treatment on BPSD (2.88, 2.92, 2.77, 3.15), effectiveness of integrative care on BPSD (2.78, 2.79, 2.75, 2.85), and safety of integrative care on BPSD (2.83, 2.84, 2.82, 2.85). Regarding the BPSD assessment method (p = 0.0134), there was a significant difference between the groups in the Chi-square test (Table 3). … Most scores on questions about the need for the contents were between 4 (agree) and 5 (strongly agree) on a 1 to 5 Likert scale: knowledge of BPSD (definition, types, assessment) (4.09 in the total sample, 4.17 in the KM hospital setting, 4.02 in the convalescent hospital setting, 4 in the public health center setting), evidence of WM treatment on BPSD (3.95, 4.07, 3.83, 3.91), evidence of KM treatment on BPSD (4.19, 4.24, 4.10, 4.36), evidence of integrative care on BPSD (4.09, 4.12, 4.02, 4.27), criteria for KM treatment referral (4.01, 4.03, 3.92, 4.36), standardized referral form for KM treatment (3.94, 3.98, 3.85, 4.18), standardized reply form of KM treatment (4.01, 4, 3.98, 4.18), and scenario of integrative medicine for BPSD (4.03, 4.10, 4.02, 3.73). There was no significant difference among the three groups in the Chi-square test (all p > 0.05) (Table 4).” (Please see page 11, red words)

Comment 3:

3)Discussion: Please, compare the findings of your work with similar survey carried out in other countries, e.g. Europe and USA

Response 3:           

Thank you for your comments. Although studies similar to this survey were rare, we added a section reviewing existing studies.

“4.3. Results of previous studies

Worldwide surveys focused on integrative medicine for BPSD have not been conducted; however, some related studies can be referenced. A US cross-sectional survey in 2007 revealed that 62.9% of community-dwelling older adults reported using one or more CAM modalities [23]. Similar results were obtained in a German cross-sectional survey in 2014, confirming that 61.3% of the elderly aged ≥ 70 years used CAM modalities [24]. When the purpose is limited to dementia, the use rate is estimated to be 18.4%, but the frequency of CAM use remains high [25]. The use of CAM in dementia patients and their caregivers is widespread, and they have an overall positive perception of the use of CAM [26]. Moreover, a qualitative study conducted on the German elderly found that most participants welcomed the incorporation of CAM into their health care [27]. However, few studies have examined the knowledge and attitudes of clinicians regarding the use of integrative medicine for dementia management. Moreover, as many CAM users do not inform their general practitioners of their CAM use [23,24], efforts to improve the knowledge and attitudes of integrative medicine from the perspective of healthcare providers in the provision of geriatric care, including dementia, are required.” (Please see page 13, red words)

Round 2

Reviewer 1 Report

The ms has improved. In my opinion it can be accepted.